# Unlocking the Future: Pluripotent Stem Cell-Based Lung Repair

**DOI:** 10.3390/cells13070635

**Published:** 2024-04-05

**Authors:** Tobias Goecke, Fabio Ius, Arjang Ruhparwar, Ulrich Martin

**Affiliations:** 1Leibniz Research Laboratories for Biotechnology and Artificial Organs, Lower Saxony Center for Biomedical Engineering, Implant Research and Development /Department of Cardiothoracic, Transplantation and Vascular Surgery, Hannover Medical School, Carl-Neuberg-Str. 1, 30625 Hannover, Germany; ius.fabio@mh-hannover.de (F.I.); ruhparwar.arjang@mh-hannover.de (A.R.); 2REBIRTH-Research Center for Translational and Regenerative Medicine, Hannover Medical School, Carl-Neuberg-Str. 1, 30625 Hannover, Germany; 3Biomedical Research in End-stage and Obstructive Lung Disease (BREATH), Member of the German Center for Lung Research (DZL), Hannover Medical School, Carl-Neuberg-Str. 1, 30625 Hannover, Germany

**Keywords:** pluripotent stem cells, hiPSC, lung, respiratory, pulmonary, differentiation, cell therapy

## Abstract

The human respiratory system is susceptible to a variety of diseases, ranging from chronic obstructive pulmonary disease (COPD) and pulmonary fibrosis to acute respiratory distress syndrome (ARDS). Today, lung diseases represent one of the major challenges to the health care sector and represent one of the leading causes of death worldwide. Current treatment options often focus on managing symptoms rather than addressing the underlying cause of the disease. The limitations of conventional therapies highlight the urgent clinical need for innovative solutions capable of repairing damaged lung tissue at a fundamental level. Pluripotent stem cell technologies have now reached clinical maturity and hold immense potential to revolutionize the landscape of lung repair and regenerative medicine. Meanwhile, human embryonic (HESCs) and human-induced pluripotent stem cells (hiPSCs) can be coaxed to differentiate into lung-specific cell types such as bronchial and alveolar epithelial cells, or pulmonary endothelial cells. This holds the promise of regenerating damaged lung tissue and restoring normal respiratory function. While methods for targeted genetic engineering of hPSCs and lung cell differentiation have substantially advanced, the required GMP-grade clinical-scale production technologies as well as the development of suitable preclinical animal models and cell application strategies are less advanced. This review provides an overview of current perspectives on PSC-based therapies for lung repair, explores key advances, and envisions future directions in this dynamic field.

## 1. Introduction

Many severe lung diseases cannot be adequately treated with existing drugs and therapies. The technology of extracorporeal membrane oxygenation (ECMO) has made great progress in recent years and can compensate for extensive loss of lung function for weeks or even months. However, although “Awake ECMO” is now available in hospitals, the available systems can only be used for in-patients and are not suitable for long-term use. In addition, unlike cardiac support systems, existing membrane oxygenators cannot be implanted. In many cases, lung transplantation remains the last therapeutic option, but this is severely limited by the far too small number of donor organs.

Cell therapy approaches may offer new therapeutic options in the future. Unfortunately, it is difficult to isolate or to expand a sufficient number of therapeutically relevant lung cells from a healthy lung, and impossible from a severely damaged one. Treatment with autologous adult mesenchymal stem cells can at best achieve immunomodulatory or limited pro-regenerative effects via paracrine factors. Pluripotent stem cells, and in particular induced pluripotent stem cells, thus appear to be the most promising cell source for cellular therapies of lung diseases.

However, while there has been impressive progress in hPSC technologies in general, and also significant progress has been made in respiratory specification of hPSC derivatives, the current state of hPSC-based therapies for lung repair is less advanced than for other organ systems such as eye, pancreas or heart, in particular due to the multi-stage development of the lung, the requirement for a plethora of pulmonary cell lineages, and the complex architecture of native lung tissue.

## 2. Pathogenesis of Life-Threatening Respiratory Diseases, Current Therapeutic Limitations and Disease-Specific Strategies for iPSC-Based Therapies

### 2.1. Pulmonary Fibrosis (PF)

PF is a chronic, progressive, and incurable interstitial lung disease characterized by abnormal accumulation of fibrotic tissue constituents, inducing alveolar histoarchitecture destruction and resulting in respiratory failure [1,2]. The pathogenesis of PF involves an interplay of genetic predisposition, environmental factors, recurrent microinjuries of a predisposed alveolar epithelium, and aberrant wound healing responses characterized by excessive collagen deposition [3,4]. Several risk factors and fibrogenic triggers could be identified to trigger PF, like smoking, virus infections, radiation, autoimmune reactions, aging, genetic predisposition, and environmental noxae [1,2,5,6,7]. PF is separated into fibrotic lung diseases with known causes, such as sarcoidosis, pneumoconiosis and chronic hypersensitivity, familial PF with known underlying mutations, and those with unknown etiology, termed idiopathic PF (IPF), which represents the most common form of lung fibrosis [1,5,8].

Currently, PF treatment is limited to a small number of drugs and supplemental oxygen [1]. Administration of corticosteroids and immunosuppressants aims to modulate inflammatory activity. However, these approaches have shown limited efficacy in controlling fibrotic processes, and their use is often associated with significant side effects. Recent advances have introduced two antifibrotic drugs, pirfenidone and nintedanib, for PF treatment [1,4]. They target pathways involved in fibrosis by acting on growth factors, growth factor receptors, interleukins, and TNF-α [1,7]. While these drugs have shown some individually differing success in slowing down disease progression, they do not provide any curative therapy [2,5,8]. Moreover, their use is often hampered by tolerability issues due to side effects or by financial constraints [4,5]. For advanced PF, lung transplantation often remains the only therapeutic option [1,8].

The main challenge in developing novel therapeutic options is the incomplete understanding of molecular and cellular mechanisms driving PF [4,9]. It is assumed that senescence of AT2 cells combined with recurrent injuries leads to the destruction of the alveolar epithelium by triggering immune responses and aberrant wound healing processes failing alveolar re-epithelialization and repair [1,2,5]. Activated alveolar cells as well as senescent AT2 cells that show a senescence associated secretory phenotype (SASP) release cytokines and growth factors that promote recruitment, proliferation and differentiation of fibroblasts and myofibroblasts secreting collagen, which ends up in excessive connective tissue deposition [5]. Mutations in genes related to cellular senescence, such as *TERT*, *TERC*, *PARN* and *RTEL1*, are associated with an increased risk of sporadic IPF [2,5,6].

However, targeting those aspects of PF pathogenesis may require a combination of therapeutic strategies with different acting mechanisms [4], including application of senolytics and cell therapy-based approaches that may allow a direct repair or replacement of dysfunctional AT2 cells as one key feature of PF pathogenesis and thus might help in resolving the devastating effects of PF [1,5,8].

### 2.2. Chronic Obstructive Pulmonary Disease (COPD)

COPD is characterized by progressive alveolar tissue destruction and defective repair mechanisms, finally leading to lung emphysema formation [10,11,12,13]. The most common symptoms are chronic cough, excessive expectoration, and exertional dyspnea [11,13]. COPD has been classically defined by the two main pathologies: (i) Emphysema with parenchymal destruction and loss of alveolar septa and (ii) chronic bronchitis consisting of chronic bronchial inflammation [12,13,14,15]. Due to its huge incidence with more than 65 million patients worldwide, COPD has become a global epidemic and now represents the third leading cause of mortality worldwide [10,11,12,14]. As age represents one of the main risk factors for COPD, current demographic changes will further increase the incidence of this disease [12,14].

COPD is triggered by chronic inflammation, an imbalance of the activity of proteases and antiproteases, and oxidative stress [10,11]. Underlying cellular and molecular determinants are not yet fully understood in detail [15]. As known today, exposure to noxious agents, mostly represented by cigarette smoke, induces an enhanced innate immune response by stimulating epithelial cells to produce reactive oxygen species, which leads to the recruitment of neutrophils, eosinophils and activated macrophages in the alveolar walls, excessive mucus secretion, and further epithelial cell activation [10,11,12,13]. Subsequent adaptive immune response is mediated by lymphocytes, which maintain chronic inflammation processes [10,11,13]. Involved neutrophils, macrophages and cytotoxic T lymphocytes are primed to release proteases and to inactivate antiproteases, leading to unregulated proteolytic degradation of extracellular matrix components [10,11,12,13]. This chronic airway inflammation, combined with defective repair mechanisms, results in progressive alveolar destruction, ultimately resulting in lung emphysema development [10,11,12,13,14]. Moreover, endothelial dysfunction of the alveolar capillary system as well as cell aging and cellular senescence contribute to the pathogenesis and progression of COPD [13].

Next to preventative steps like smoking cessation and supportive methods like oxygen supplementation, only a few pharmaceutic therapeutic options are currently available, targeting solely a symptomatic treatment of COPD [12,14]. All of these substances, applied alone or in combination, fail to curatively address the underlying pathomechanisms of COPD, and thus, neither disease-related decline in lung function nor mortality can be influenced sustainably at present [10,11,12]. In end-stage respiratory failure due to severe lung emphysema, lung transplantation often represents the only remaining therapeutic option [12]. In contrast, it is likely that an application of regenerative therapies that may complement existing pharmacological treatment will be rather restricted to earlier disease stages, as the terminal loss of the alveolar structure is considered to be irreversible [11,12]. In particular, senolytic treatment, followed by replacement of senescent cells by juvenile hPSC-derivatives, may be a promising concept that addresses one of the underlying key mechanisms of COPD. In combination with agents that modulate specific inflammatory pathways, with novel bronchodilators with improved efficacy and reduced side effects, and with novel mucolytic agents and antimicrobial therapies, such approaches may revolutionize the management of COPD [10,11,12,13,14,15].

### 2.3. Pulmonary Arterial Hypertension (PAH)

PAH constitutes a complex and multifactorial pulmonary arterial (PA) vasculopathy with structural changes including occlusive vascular remodeling driven by excessive vascular proliferation and inflammation [16,17]. The pathogenesis of PAH is multifaceted, and the underlying mechanisms are still not fully elucidated [16,18]. PA blood flow obstruction and thus an increase in PVR with subsequent right ventricular remodeling and finally right heart failure are the consequences [16,18,19,20], leading to an average lifespan of 5–7 years after diagnosis [17,21].

Today some pathological conditions and gene defects such as reduced expression of *BMPR2* or autosomal dominant mutations in the *BMPR2* gene in hereditary PAH are known to favor disease development [19]. Mutated ECs and resulting impaired signaling with SMCs as well as endothelial to mesenchymal transition and general loss of ECs in the PA vasculature and inflammatory cell infiltration are assumed to cause obliterative growth, neointimal hypertrophy, and remodeling in the pulmonary vasculature [16,17,18].

The pivotal role of endothelial dysfunction in PAH underscores the need for therapies that target this fundamental aspect of pathogenesis [17,19]. Only a few drugs available today have demonstrated efficacy in reducing vasoconstriction and providing symptomatic relief for PAH patients as well as improving their overall clinical outcomes, but none of them emerged to be effective in inhibiting vascular remodeling and fibrosis and thus PAH progression [22]. The most recent insights in recognizing the role of inflammation in PAH have opened new ways for immunomodulatory therapies whereby understanding the immune interactions within the pulmonary vasculature will reveal novel targets to address the immune dysregulation contributing to PAH pathogenesis [22,23,24]. Recent advances have yielded innovative agents targeting impaired signaling pathways such as soluble guanylate cyclase stimulators and tyrosine kinase inhibitors [20]. 

However, these therapeutics are still not able to cure or reverse PAH by addressing its molecular mechanisms and, therefore, only possess the potential for symptomatic relief and slowing down disease progression under acceptance of severe adverse effects [17,20]. Thus, the development of novel regenerative therapeutic strategies that target pulmonary vascular ECs are urgently required, and replacement of ECs carrying PAH-causing mutations by genetically corrected or healthy WT hiPS-ECs could offer hope for patients struggling with this deleterious disease [17,18,21].

### 2.4. Cystic Fibrosis Lung Disease (CF)

CF is an autosomal recessive inherited disease caused by mutations in the CF transmembrane conductance regulator (*CFTR*) gene and represents the most common life-limiting monogenic disease in Caucasian populations, affecting approximately 90,000 individuals worldwide [25,26,27]. CF is a progressive genetic multi-system disease [25,27,28]. However, morbidity and mortality of CF patients are mainly caused by respiratory impairments [25,28,29]. CF lung disease is characterized by chronic airway inflammation, mucus hypersecretion, and recurrent infections, ultimately leading to progressive lung damage and destruction [28,29].

Although CF is caused by mutations in a single gene, over 2000 genetic variants have been identified to date [25,27]. In the airway epithelia of the lung, *CFTR* dysfunction disrupts chloride and bicarbonate transport, leading to an accumulation of highly viscous mucus, impaired mucociliary clearance, and an impaired activity of antimicrobial enzymes. Taken together, these factors create a pro-inflammatory microenvironment conducive to persistent bacterial colonization [25,26,27,28], initiating a vicious cycle in the small airways represented by airway obstruction, excessive inflammation, and chronic infection [25,27,28,29]. The latter triggers an exaggerated inflammatory response, perpetuating chronic inflammation, consecutive structural tissue damage, progressive airway remodeling, and steadily decreasing lung function [25,26,29].

Symptomatic therapies such as airway clearance, mucus thinning agents, antibiotics, and anti-inflammatories remain crucial for the management of CF airways and can increase the life expectancy of CF patients as well as reduce both the frequency and severity of pulmonary exacerbations [25,27,28]. In addition, management of persistent inflammation and infection requires anti-inflammatory and anti-infective therapies that may support *CFTR* modulator therapy [29]. Emerging therapies are aimed at improving mucociliary clearance by reducing mucus viscosity. Most importantly, the advent of small molecule *CFTR* modulators—especially the triple combination of ivacaftor, lumacaftor and tezacaftor—can be considered as a breakthrough and has proven to be promising in improving chloride transport and symptoms [25,28,30]. *CFTR* modulator therapies lead to an improvement in lung function and quality of life and thus make lung transplantation, the only remaining life-saving therapeutic option in former times, largely superfluous [25,28,29]. However, the triple combination does not work for all type of mutations. The variability in therapeutic response, yet still missing long-term efficacy data, adverse effects and challenges associated with specific mutations are still causative for the varying prognosis of CF patients and underscore the need for further advances in this area [28,29,31].

As there is still no curative treatment for CF available today, CF patients are faced with a limited life expectancy as well as a reduced quality of life due to the occurrence of severe pulmonary complications and the lifelong intake of cost-intensive drugs that are accompanied by considerable side effects [25]. Moreover, the heterogeneity of CF mutations calls for personalized curative approaches aiming at repair or replacement of airway basal cells carrying the underlying genetic defect [28,32]. One challenging aspect of such therapeutic concepts is that basal cells, which are responsible for continuous airway regeneration, are located below the apical ciliated and goblet cells. Functionally correcting solely the apical ciliated epithelia would limit therapeutic effects to just several days to weeks; therefore, novel approaches have to be developed to replace existing mutated basal cells.

### 2.5. Other Genetic Lung Diseases

Other lung disorders rooted in inherited genetic mutations present unique challenges both in understanding their pathogenesis and in developing effective therapeutic interventions. A variety of genetic mutations contribute to the pathogenesis of different genetic lung diseases, and only some of the most important rare lung diseases are described below.

Alpha-1 Antitrypsin Deficiency (AATD), a rare hereditary disorder affecting the production of alpha-1 antitrypsin (AAT) protein, is characterized by a deficiency of circulating AAT [33,34,35]. Ninety-five percent of severe AATD cases result from the *Glu342Lys* mutation (*Z allele*), which is found in 1 out of 25 members of the north European white population, with 1 out of 2000 individuals carrying 2 *Z alleles* and thus are homozygotes [33,34,35]. It is estimated that over 120,000 European individuals have severe AATD [35]. Lack of AAT predisposes homozygotes to early onset emphysema [33,34,35]. Behavioral interventions such as the avoidance of smoking and augmentation with pooled plasma AAT still represent the only treatment for AATD [33,34,35]. Augmentation therapy admittedly leads to raised levels of AAT and, therefore, is able to reduce inflammation, but it is not suitable to stop or reverse emphysema progression [33,35]. And thus, lung transplantation often represents the only remaining therapeutic option for AATD patients. Novel therapies aimed at curative therapeutic strategies for AATD, including corrected or functionalized induced pluripotent stem cell derivatives, offer potential opportunities for affected patients [35].

Primary Ciliary Dyskinesia (PCD) is a rare heterogeneous genetic ciliopathy characterized by abnormal ciliary structure and impaired motional function, which, in terms of the airway epithelium, results in reduced or absent mucociliary clearance and thus in recurrent respiratory tract infections [36,37]. Estimations of PCD prevalence are difficult, since diagnosis is often delayed or entirely hindered because most of the symptoms overlap with other, more common pathologies, and a gold-standard diagnostic test is not available yet [36]. The course of PCD constantly includes a progressive decline of lung function due to structural lung damage caused by recurring airway infections [36,37]. Currently, there is only symptomatic treatment, mostly based on strategies also applied for patients with CF or non-CF bronchiectasis [37]. In some cases, lung transplantation can be necessitated as the last possible therapy due to rapid disease progression and lung failure. PCD is a clinically and genetically highly heterogenic disorder, with patients showing a wide spectrum of phenotypes from mild symptoms to acute life-threatening conditions [36,37]. Causal for this is the high number of different underlying genes that can be affected in different ways by a variety of different deleterious mutations [36,37]. Hence, the development of personalized therapeutic approaches tailored to each individual patient, including induced pluripotent stem cells and genome editing technologies according to the specific genetic diagnostic, will be inevitable for future PCD management [36,37]. Similarly to CF, novel therapies restoring and normalizing ciliary function will only be possible via targeting airway basal cells that constantly regenerate the airway epithelia [37,38].

Genetic surfactant deficiencies represent another group of rare lung diseases. Pulmonary surfactant is a complex mixture of ~90% phospholipids and ~10% proteins produced by AT2 cells. It is necessary to reduce the alveolar surface tension and to prevent atelectasis. The genetics of surfactant deficiencies are complex, and most mutations that have been identified to cause surfactant deficiencies are located in the genes *SFTPA1*, *SFTPA2*, *SFTPB*, *SFTPC*, and *SFTPD* encoding the surfactant proteins A, B, C, and D, in ABCA3 (a protein required for surfactant packaging and secretion), and in the transcription factor NKX2, which is involved in regulation of surfactant protein expression [39].

Some mutations are autosomal dominant, and others autosomal recessive. While homozygous mutants in SP-A seem to be incompatible with extrauterine life, the most severe genetic surfactant deficiencies are homozygous SP-B and ABCA3 deficiencies, as well as heterozygous NKX2 deficiencies, which are often lethal in early childhood. Affected patients can be rescued by lung transplantation only. In addition to personalized gene therapies, administration of genetically healthy or gene-corrected hPSC-derived AT2 cells may represent a promising new therapeutic option.

Pulmonary alveolar proteinosis represents another rare lung disease [40]. To date, whole lung lavage (WLL) remains the gold standard treatment for PAP syndrome. However, in patients with hereditary PAP with underlying GM-CSF receptor mutations, ex vivo autologous hematopoietic stem-cell gene therapy and transplantation of autologous iPSC-derived ex vivo gene-corrected macrophages directly into the lungs are promising approaches [41].

In general, genetic lung diseases, each with a distinct pathogenesis, require tailored therapeutic approaches that address the underlying genetic defects. The clinical need for effective therapies in genetic lung diseases requires a collaborative effort to bridge the gap between genetic understanding and therapeutic advances offering hope to individuals affected by these complex disorders.

## 3. Current State and Limitations of Human Pluripotent Stem Cell Technologies

Human pluripotent stem cell (hPSC) technologies have generated immense excitement in the field of regenerative medicine with the potential to revolutionize lung repair and the treatment of respiratory diseases. Meanwhile, many hurdles and limitations for production of clinically applicable iPSC derivatives have already been overcome. Transgene-free iPSCs can be efficiently derived under fully defined GMP-compliant conditions from easily accessible cell sources such as blood [42], and highly efficient protocols for site-specific (and thus relatively safe) genetic engineering by homologous recombination based on engineered nucleases, in particular the CRISPR/Cas9 system, have been successfully established [43]. Utilizing these novel tools, it has been possible to develop protocols for footprintless gene editing without the need for antibiotic selection [44]. Such techniques are of key importance for preclinical animal studies and future cellular therapies, as they allow, e.g., the correction of disease-related mutations or the well-defined expression of reporter transgenes that facilitate the monitoring of graft survival, functional integration and distribution in small and large animals [45]. Moreover, the targeted introduction of transgenes into safe-harbor sites [46] is possible, enabling for instance, overexpression of therapeutic transgenes without oncogenic potential, or allowing the introduction of an inducible suicide system for targeted killing of the cellular graft in case of tumor formation [47,48].

iPSCs can now be expanded in scalable suspension culture [49], and large numbers of human iPSCs can be produced in fully controlled bioreactors [50]. Moreover, for some cell lineages, sequential inhibition and activation of molecular differentiation pathways has allowed a targeted, more robust, and efficient scalable differentiation of human ESCs and iPSCs for the first time [51]. In some cases, all required recombinant proteins could be replaced by small molecules, which has greatly improved the robustness of such differentiation protocols [52,53]. The introduction of small molecules as activators or inhibitors of molecular key pathways has facilitated the development of scalable protocols that are relatively inexpensive and more robust. Such protocols have been developed for various cell types, including cardiomyocytes [53], ECs [51], macrophages [54], endodermal progenitors [55], hepatocyte-like cells [56], and neural derivatives [57], but not for any respiratory epithelia so far.

While specific hPSC derivatives can, meanwhile, be produced at large scale and in high purity, the maturity of these cells is often a matter of debate, e.g., in the case of cardiomyocytes [58] or hepatocytes [59]. Although immaturity is certainly a major limitation when such cells are used to model adult human disease in a dish, immaturity allowing for more efficient functional integration after transplantation may even pose a benefit in the case of therapeutic applications since engrafted cells are expected to further mature in vivo over time.

The risk of formation of teratoma, a type of tumor containing cells from all three germ layers arising from undifferentiated pluripotent cells, remains a concern with hPSC-based therapies as long as no sufficient purity can be guaranteed. However, in view of the continuous improvement of differentiation protocols and the availability of proper enrichment, purification, and analysis technologies, it should be possible to exclude the presence of remaining undifferentiated cells within the differentiated cell product. Therefore, the risk of teratoma formation seems to be very low if highly enriched cell products are used.

In addition to the assessment and reduction of teratoma risk, research on safety issues also addresses the appearance of genetic and epigenetic abnormalities [60], which is probably a more critical aspect for future clinical application [61]. Much effort has already been invested to analyze critical chromosomal abnormalities and to identify critical genes such as *p53* [62] and specific mutations that may cause malignant transformation. It is also necessary to carefully investigate to what extent mutations are already present in the source cells, are generated during reprogramming, or are enriched during subsequent hiPSC expansion, which is mandatory for many therapeutic applications [63]. Therefore, various studies investigated the origin of small- and large-scale genetic aberrations in pluripotent stem cells and their progeny [64,65]. Certainly, these questions, suitable quality control measures, and exclusion criteria are already considered [66,67,68].

Aside from these aspects, also proper therapeutic application approaches that result in efficient functional integration of the transplanted cells are still a major limitation and have not been sufficiently addressed so far. For example, recent experiments in animals, including non-human primates [69], revealed that the majority of cardiomyocytes injected to achieve repair of infected hearts are already lost during application via the injection channel and the venous route, and more research is required to investigate distribution, survival and adverse effects of these cells.

As hPSC technologies advance, collaborative efforts between scientists, clinicians, and regulators will be critical to address these challenges and pave the way for transformative regenerative therapies in lung repair.

## 4. Targeted Production of PSC-Derived Cell Lineages Relevant for Respiratory Diseases

Depending on the disease, different cell lineages may be required for cellular therapy of the lung (Table 1). While in PAH, ECs [70] and SMCs [71] represent the affected and potentially disease-causing cells, different epithelial cell types and resident epithelial stem/progenitor cells, as well as ECs and SMCs, are discussed as disease-causing respiratory cell lineages in COPD [72,73]. Whereas cells of the proximal airways, including ciliated airway epithelium, goblet cells, and basal cells that allow regeneration of these cells, are considered as disease-causing/target cell types in cystic fibrosis [74] and primary ciliary dysplasia [36,75], type II alveolar epithelial cells seem to play a decisive role not only in inborn surfactant deficiencies [39] but also in pulmonary fibrosis [76]. Finally, macrophages represent another relevant cell type, e.g., for treatment of pulmonary alveolar proteinosis (PAP) [77,78] or bacterial airway infections [79].

The differentiation of hPSCs into lung-specific cell types, such as respiratory epithelial cells, ECs and SMCs, but also macrophages and lung mesenchyme, has advanced significantly. Innovative differentiation protocols using growth factors, small molecules and signaling pathways have enabled the targeted generation of individual cell populations critical for lung repair, although in some cases not yet available as highly enriched cell populations.

In the case of ECs [51,80,81], SMCs [82] and lung-specific mesenchyme [83], protocols are now in place that allow the efficient generation of functional cells, although unpublished data show that the capacity of ECs for further expansion is rather limited for reasons unknown so far.

Especially in the case of ECs, it is still being discussed as to whether it is necessary to achieve specification into endothelial subtypes, e.g., (respiratory) microvascular ECs [84], or whether the generated ECs express sufficient plasticity to appropriately adapt to the given niche in vivo. Pericytes are considered important to support vascular sprouting. Although the identification, enrichment, and characterization of these cells are still difficult due to the lack of truly specific pericyte markers, there are now protocols for the targeted generation of pericytes [81,85,86]. SMCs, as the third important vascular cell type, can also be differentiated together with ECs from iPSCs via a common vascular progenitor [87].

iPSC-derived macrophages represent a promising cell type for treatment of PAP [77] and pulmonary bacterial infections, especially if caused by multidrug-resistant pathogens [79]. iPS-derived macrophages can already be produced under defined conditions in scalable suspension culture applying spinner flasks or fully controlled bioreactor systems [79].

For most lung diseases, however, production of respiratory epithelial cells or their progenitors presumably represents the key for cellular therapies. Although first attempts to generate alveolar [88] and airway [89] epithelia from murine embryonic stem cells showed limited success already two decades ago, major efforts were required to translate those early findings into more defined culture conditions [90,91]. While initially, differentiation in embryoid bodies or under co-culture conditions was often applied [88,89], insights especially from mouse development enabled a stepwise approach for targeted differentiation of hPSCs via endoderm, foregut endoderm, and early lung progenitors into proximal and distal lung epithelia [92].

Here, the work of Kotton et al. [93] and Snoeck et al. [94,95] provided the crucial basis for all further developments. Subsequent work finally led to the development of chemically defined protocols for targeted differentiation of hPSCs into lung progenitors [95,96] and more mature airway epithelia [97] (Figure 1) including ciliated cells, goblet cells [98], and basal cells [99,100], which are key for continuous regeneration of the airway epithelium. Meanwhile, differentiation into type II alveolar epithelial cells [101,102] such as the surfactant-producing cells of the alveoli that also fulfill immunological tasks and a second important function as alveolar stem cells that give rise to type I alveolar epithelial cells [103] is also possible. Importantly, now there are also protocols in place that prevent immediate differentiation of cultured AT2 cells into AT1 cells accompanied by loss of stem cell function, and in vitro expansion of hiPSC-derived AT2 cells seems possible [104,105].

Interestingly, it turned out that robust protocols for stepwise targeted differentiation of hPSCs via sequential activation and inhibition of signaling pathways are easier to establish for human cells than for mouse cells. The most probable reason for this finding is that lung development in the mouse requires only 19 days as compared to 9 months in humans. Thus, proper timing of the activation/inhibition of signaling cascades is more difficult in murine cells, even more so because every PSC line behaves differently, including the proliferation rate. Nevertheless, recent work also demonstrated the derivation of murine PSC-derived basal cells [106] and alveolar epithelia [107].

At this point, however, there are still no scalable protocols available that allow production of defined epithelial cell populations in suspension culture, cells cannot be differentiated into highly enriched target cell populations just by sequential inhibition or activation of signaling cascades, and typically laborious cell sorting steps are required to produce the desired cell type at sufficient purity. And finally, no GMP-compliant protocols, that are required for clinical application, have been developed. As we navigate this frontier, the integration of innovative differentiation protocols, 3D culture systems and bioprocess technologies is promising for the generation of functional cells relevant to respiratory regeneration. Collaborative efforts across disciplines are essential to overcome remaining challenges and to push hPSC-based therapies closer to the clinical treatment of lung diseases.

## 5. Preclinical Models for Evaluating Novel Cell Therapies in Lung Diseases

As the search for effective cellular therapies for lung diseases intensifies, the use of reliable animal models for preclinical evaluation is crucial. Suitable models should reflect human physiology and tissue architecture as closely as possible. For instance, the anatomy and histology of airways in small rodents are different from humans especially with regard to the distribution of basal cells and club cells [108,109]. Also, the model should mirror underlying pathomechanisms and the phenotype of the respective disease. Finally, therapy-relevant issues and experimental requirements also have to be considered. For example, immunodeficient animals have to be used to enable engraftment and survival of transplanted human cells, and the size of the animals should be sufficient to facilitate surgical procedures.

If no immunodeficient animals are available, e.g., in the case of genetic mouse models, also murine PSC-derivatives can be transplanted, which requires major efforts to develop suitable differentiation protocols that allow production of a relatively high number of sufficiently pure cells for transplantation studies [106,107]. After a first study by Rosen et al. applying primary lung cells [110], Herriges et al. and Ma et al. were able to demonstrate robust engraftment of PSC-derived airway and alveolar epithelia for the first time [106,107].

There are several drug-induced models for pulmonary fibrosis (PF). The most common animal model in pulmonary fibrosis research is the bleomycin-induced fibrosis in mice and rats. This model recapitulates key features of human pulmonary fibrosis and has been instrumental in the evaluation of antifibrotic agents. Bleomycin (BLM) is a cytostatic glycopeptide that is usually applied via intratracheal instillation and induces acute lung injury, with the first signs of pulmonary fibrosis becoming visible by approx. day 3 after treatment [111]. The precise mechanisms underlying BLM-induced pulmonary fibrosis have not been established. Through activated BLM-induced DNA cleavage, free radical formation, and deoxynucleotide oxidizing reaction, BLM induces alveolar epithelial injury and apparently also a senescent phenotype [112]. As a result of a dysfunctional repair response, BLM produces pulmonary fibrosis [113].

Other drugs frequently used to induce pulmonary fibrosis are amiodarone and methotrexate [113]. Amiodarone mainly induces intracellular phospholipid accumulation (phospholipidosis) in AT2 cells and alveolar macrophages. Methotrexate significantly promotes the epithelial–mesenchymal transition (EMT) process of AT2 cells [113]. While all compounds seem to induce fibrosis via AT2 cells, none of the models addresses genetic causes of familial and idiopathic fibrosis. Thus, recent models were not able to directly investigate the effect of engrafted gene-corrected or WT iPSC derivatives on fibrosis caused by the mutated endogenous AT2 cells [114].

Most PF-causing mutations affect surfactant-related genes or genes of components of the telomerase complex, others cell cycle-related genes, mucus genes, or genes with immune functions [115]. Accordingly, most genetic animal models are mouse models carrying knock-outs of or mutations in relevant genes. Importantly, the developed mouse strains have not been bred on an immunocompromised background. Therefore, application of human gene-corrected or WT iPSC derivatives would require back-crossing with immunocompromised mouse strains, or potentially administration of high levels of immunosuppression to prevent xenogeneic rejection of the instilled human cells.

Until 2014, preclinical animal experiments related to cell-based treatment of PF were restricted to mesenchymal stem cells, stem cell-derived exosomes or secretome of PSCs [116], which were reported to show beneficial effects on bleomycin-induced lung injury and fibrosis [1,117].

Thereafter, first experimental studies applying hiPSC-derived AT2 cell preparations [118] in SCID mouse or immunosuppressed rat bleomycin models were also conducted [118]. Although apparently no highly enriched AT2 cell preparations were applied, and although no convincing evidence for engraftment of hPSC-derived AT2 cells was shown, the treatment obviously resulted in significant reduction of bleomycin injury and potentially fibrosis [118,119], raising hope for an effective and safe treatment of PF via instillation of iPSC-AT2 cells.

When choosing a suitable animal model in COPD research, a critical question is always to what extent lung anatomy and histology of the animals resembles the human lung. While mice and rats are easy to handle, inexpensive, and offer the opportunity to use genetically modified animals, development of COPD features is strain-dependent. Furthermore mice do not have submucosal glands, and distribution of basal cells in the airways is substantially different from that in humans [108,109]. Other species used in COPD research are guinea pigs and dogs, but also ferrets and monkeys, which show a physiology and cell distribution similar to that in humans. In monkeys, the immune system is also very similar [120].

Clearly, all established animal models reflect human COPD pathologies to a limited extent and are not able to fully recapitulate the entire spectrum of clinical COPD [120]. Remarkably, most animal models of emphysema do not truly mimic the human disease, but rather reflect airspace enlargement due to either loss of collagen/elastin or tobacco smoke-induced insults. The most common animal models in COPD are based on treatment with elastase, which induces emphysema, or on a long-term treatment with tobacco smoke or smoke extracts. The advantages and limitations of the different models, that have been used have been described in detail by Upadhyay et al. [120]. One recent study applied primary pulmonary ECs to reverse emphysema in an elastase-induced mouse model. Although no PSC-derivatives were applied, this study raises hope that healthy PSC-derived ECs may be therapeutically useful to tackle emphysema [15]. Another study suggested an approach to enable relatively efficient engraftment of healthy basal cells [106], a second cell type that seems to be key for development of COPD.

Different animal models are currently used in PAH. Widely used is the monocrotaline (MCT) model. While mice are unable to convert monocrotaline in the liver to its pneumotoxic metabolite MCT, administration of monocrotaline in rats closely mimics the phenotype of PAH [121]. On the other hand, underlying pathomechanisms in human PAH are not well recapitulated in this model. The Sugen/hypoxia model [122], which is also used in mice despite being apparently more reliable in rats, addresses the mechanisms of human PAH induction more closely. However, this model also does not mimic the contribution of (hereditary) genetic alterations, in particular mutations in the *BMPR2*. This is addressed in various genetic mouse and rat models [123], the more recent ones being generated using gene editing technology [124,125]. For more detailed information about animal models in PAH, the review of Boucherat et al. is recommended [123].

The vast majority of cellular therapies conducted so far applied MSCs, MSC-derived exosomes or EPCs aiming at enhanced vascular repair and inhibition of endothelial–mesenchymal transition, inflammation, and apoptosis. The first application of mouse iPSCs was reported in 2016 [126]. While undifferentiated iPSCs and their conditioned medium were used in this study to investigate paracrine effects, there has been no experimental study to date that aimed to replace mutant ECs by injection of iPSC-derived EPCs or ECs.

Currently, three important animal species are used in CF research. In mice, ferrets and pigs, CF transmembrane conductance regulator (*CFTR*) knockout models and also models with specific mutations introduced into the *CFTR* gene are available. In 2014, 2016 and 2018 also, *CFTR* KOs were generated in rats, rabbits, and sheep [74], but limited data are available for these relatively new models.

Mice poorly reflect the anatomy, histology, and cell composition of the human airways, and *CFTR* knockout models in mice only mimic the human CF phenotype to a limited extent, especially regarding the lung pathology. Pigs and ferrets much more closely mimic CF pathology [74]. Although mimicking CF pathology and pathomechanisms works quite well in these models, there are major limitations of the models when it comes to cellular approaches aiming at the functional restoration of the airway epithelium. Importantly, there is no model in immunocompromised animals yet that enables engraftment of xenogeneic human cells, and substantial levels of immunosuppression would probably be required to achieve survival of transplanted hPSC derivatives, a treatment that may well affect disease phenotype and experimental outcome. Unfortunately, pluripotent stem cells of high quality that could supersede pharmacological immunosuppression are still unavailable for ferret or pig. On the other hand, it seems that optimized protocols of lung injury via irradiation and chemical treatments with napthalene [110] or polidocanol [106] prior to cell application may pave the way for efficient replacement of mutated endogenous basal cells through allogeneic WT or genetically corrected autologous PSC-derived basal cells.

For cell-based treatment of genetic surfactant deficiencies, replacement of mutated AT2 cells by gene-corrected autologous iPSC-derived AT2 cells would most likely be the method of choice. In contrast to basal cells, AT2 cells are directly accessible in the alveoli, and engraftment of murine iPS-derived AT2 cells has already been demonstrated [107].

Closer to clinical application, however, may be the treatment of PAP and bacterial infections via iPSC-derived macrophages that can already be produced at clinical scale under generally GMP-compliant conditions [54]. In mouse models, hiPSC-derived macrophages already have been applied successfully to treat PAP [78] and *Pseudomonas* infections of the lung as well [79].

In summary, the synergy between cell production technologies, novel therapeutic concepts, and advanced animal models is shaping the landscape of iPSC-based treatment of lung diseases. The applied models should not only recapitulate disease pathologies but also serve as crucial platforms for the evaluation of emerging therapies. Safety certainly remains a crucial concern, especially with pluripotent stem cells. Ensuring that transplanted cells do not contribute to tumor formation or exhibit uncontrolled differentiation can be addressed in animal models and should at least allow a significant limitation of those risks prior to clinical application.

## 6. Perspectives for Clinical Trials

The translation of pluripotent stem cell (PSC)-based therapies from preclinical studies to clinical applications could herald a new era in the search for effective treatments for lung disease.

According to ClinicalTrials.gov, there are already a number of clinical studies, either terminated, ongoing or planned, that apply autologous or allogeneic mesenchymal stem cells for treatment of interstitial lung disease, bronchopulmonary dysplasia, COPD, acute respiratory distress syndrome or induced lung injury (e.g., ClinicalTrials.gov NCT02594839, NCT01919827, NCT04062136, NCT03558334, NCT03601416, NCT03873506, NCT03774537, NCT02645305, NCT04433104, NCT01758055, NCT02804945, NCT02444858). While MSCs may provide immunomodulatory or paracrine effects, other studies apply lung stem cells for treatment of interstitial lung disease, aiming to replace endogenous stem cells that are affected by the disease. In particular, airway cells that have been dedifferentiated into expandable basal cells in culture have been transplanted (e.g., ClinicalTrials.gov NCT02796781, NCT02745184, NCT04262167).

Autologous endothelial progenitor cells (EPCs) have been applied for the treatment of IPF and PAH (e.g., ClinicalTrials.gov NCT00641836, NCT00372346, NCT00257413); however, the rationale behind these trials remains unclear since the transplanted autologous cells also carry those mutations and genetic variants that may have contributed to the development of the respective disease.

In contrast, there are currently no studies listed in ClinicalTrials.gov that apply pluripotent stem cell derivatives for the treatment of lung diseases.

For the therapeutic application of functional derivatives of pluripotent stem cells, a whole series of challenges have had and still have to be overcome, and risks have to be evaluated and minimized. From an ethical, regulatory, cost, and therapeutic point of view, it is, furthermore, necessary to decide on a case-by-case basis whether therapy with allogeneic cells, derived from existing ECS or iPSC lines, or therapy with autologous cells, derived from the body’s own iPSCs, is an option.

Cellular therapies based on pluripotent stem cells (iPSCs)—for the first time—offer the replacement of mutated malfunctioning or senescent pulmonary cells that are causative for the respective disease. There is now a growing number of clinical trials in which derivatives of human ESCs or iPSCs are being used to repair other organs and treat other diseases (see https://clinicaltrials.gov/ and more specific under https://hpscreg.eu/browse/trials), but there are still no such clinical trials for the treatment of respiratory diseases. However, since protocols for cell differentiation and scalable bioprocess technologies have now been developed and substantially advanced, it can be presumed that the first clinical trials will also be initiated in the field of PSC-based lung repair.

## Figures and Tables

**Figure 1 cells-13-00635-f001:**
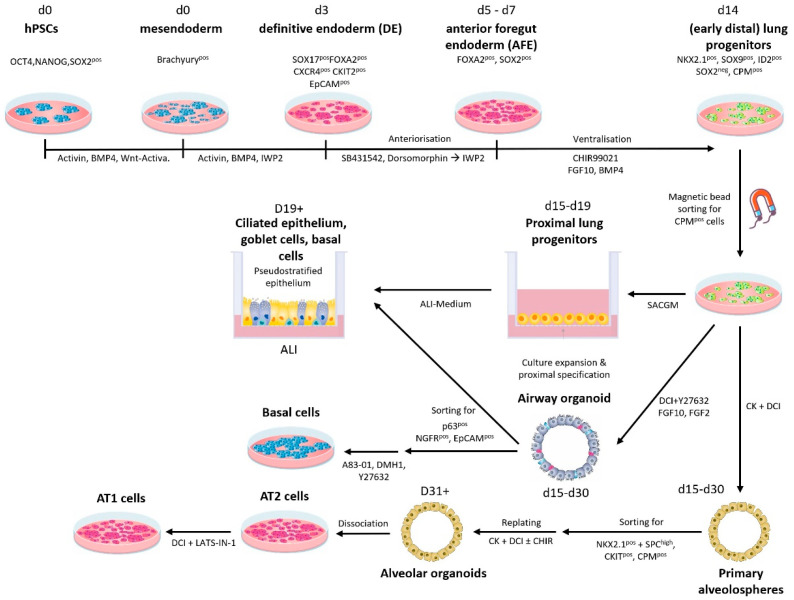
Current protocols for respiratory in vitro differentiation of hPSCs. Abbreviations:; ALI, air liquid interface; CK, CHIR (CHIR99021, *GSK-3* Inhibitor/*Wnt* Activator +keratinocyte growth factor (KGF); CPM, carboxypeptidase M; DCI, dexamethasone, 3′,5′-cyclic adenosine monophosphate (cAMP), and 3-isobutyl-1-methylxanthine (IBMX); IWP2 (*Wnt*/ß-catenin inhibitor); LATS-IN-1 (Lat1/2 kinases Inhibitor); SACGM, small airway epithelial cell growth medium; SB431542 (TGFß-R Inhibitor); Y27632 (ROCK-Inhibitor).

**Table 1 cells-13-00635-t001:** Candidate cell types for iPSC-based therapies of common and rare lung diseases.

Cell Type	ECs		Basal Cells		AT2 Cells		Macrophages	
Lung Disease	Allo	Auto	Allo	Auto	Allo	Auto	Allo	Auto
PAH	X							
COPD	X		X					
Fibrosis					X			
Cystic Fibrosis				X				
PCDs				X				
SDs						X		
PAP								X
Bacterial Inf.							X	

Abbreviations: Allo, allogeneic; auto, autologous; AT2 cells, alveolar type II epithelial cells; COPD, chronic obstructive pulmonary disease; ECs, endothelial cells; PAH, pulmonary arterial hypertension; PCD, primary ciliary dyskinesia; SDs, surfactant deficiencies; PAP, pulmonary alveolar proteinosis; Inf., infections. X = Most likely clinical applications considering the typical age of patients and the cost-benefit-ratio in respective patient groups.

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
