# Peer review of "Unlocking the Future: Pluripotent Stem Cell-Based Lung Repair"

_cells, 2024, doi:10.3390/cells13070635_

Round 1
Reviewer 1 Report
Comments and Suggestions for Authors
The review paper begins by offering an overview of the pathogenesis of various end-stage pulmonary diseases, for which many treatment options exist for symptom-relief but all lack definitive cures. The paper described the role of disease-causing cells and/or compartments for each condition and our current understanding of the pathological signalling pathways and mechanisms. For its latter part, the paper focuses on recent advances in and current limitations of PSC-based therapies. It summarizes the types of respiratory cells, such as ciliated, goblet and basal cells, and structural cells, such as endothelial and smooth muscle cells, that can be differentiate from PSCs, as well as existing issues, including safety, efficacy, engraftment efficiency and the lack of scalable protocol, that must be addressed.
Overall, the paper quotes other review papers frequently, makes statements that are rather broad and generalized. This is particularly noticeable in the first section of the paper. In fact, the detailed description of pulmonary pathology is a considerable section of the paper and while nicely summarized in this review, there have been numerous other reviews which summarize end-stage pulmonary diseases.
It is unclear what the focus of the review paper is. If the main purpose of the review paper is to summarize current perspectives of PSC-based therapies for lung repair and recent advances, it is not necessary to spend the majority of the manuscript detailing the pathogenesis of various respiratory diseases. Authors could perhaps focus on the in vitro and in vivo applications of PSCs (i.e. disease modelling). These disease describing sections can then be significantly shortened. If not, the paper should be re-titled and the abstract revised to clarify that the paper mainly focuses on recent advances in respiratory disease pathogenesis and treatment (which can include the use of PSCs for modelling or therapy).
The summarizing paragraph/sentences after each respiratory disease section is redundant. They are all essentially speaking to the fact that current treatment options for these end-stage lung diseases are far from satisfactory, novel therapeutic strategies should target newly identified pathways and mechanisms and PSC-based therapies hold great promise to cure these diseases. All these paragraphs can be combined into one and placed and the end of chapter/section 1.
Certain statements need to be explained or backed up with more specific examples. For example, page 6 line 290: “Understanding the immune interactions within the pulmonary vasculature may reveal novel targets that address the immune dysregulation that contributes to the pathogenesis of PAH.”, specific components of the immune system that can be targeted should be explicitly stated.
The paper should also avoid generalized quoting of other review papers (e.g. references 23-25) and the use of broad statements (e.g. “Moreover, for some cell lineages, sequential inhibition and activation of molecular differentiation pathways has, for the first time, allowed a targeted, more robust and efficient scalable differentiation of human ESCs and iPSCs [57].”)
Comments on the Quality of English LanguageThere are grammatical errors throughout the paper and should be corrected with special attention to sentence structure.
Reviewer 2 Report
Comments and Suggestions for Authors
In this review, the authors describe the vulnerability of the human respiratory system to diseases such as COPD, pulmonary fibrosis and ARDS, among others, emphasising the global challenge these diseases represent for the health sector, as they are one of the leading causes of death worldwide. Due to the limited efficacy of current therapies, there is a need for innovative solutions to regenerate lung damage. The authors highlight the potential of pluripotent stem cells to develop regenerative therapies, as protocols are now available to differentiate these cells into different lung cell types. However, they caution that despite advances in genetic engineering and cell differentiation, challenges persist in the development of production technologies, preclinical models, and effective cell application strategies. In summary, this review provides an overview of pluripotent stem cell-based therapies for lung repair, discussing key advances and projecting future directions in the field.
The review is well-written and easily understandable. Perhaps, it is recommended to include in the discussion some results from preclinical and clinical studies that have already been conducted with isolated primary lung cells, showing promising results. This additional information would further support the potential of regenerative therapies using differentiated pluripotent stem cells for various lung cell types.
